# The Clonal Spread and Persistence of *Campylobacter* in Danish Broiler Farms and Its Association with Human Infections

**DOI:** 10.3390/pathogens14080821

**Published:** 2025-08-19

**Authors:** Katrine Grimstrup Joensen, Gitte Sørensen, Pernille Gymoese, Louise Gade Dahl, Eva Møller Nielsen

**Affiliations:** Department of Bacteria, Parasites & Fungi, Statens Serum Institut, 2300 Copenhagen S, Denmark

**Keywords:** whole-genome sequencing, genomic surveillance, longitudinal strain persistence, inter-farm transmission, zoonosis, One Health, foodborne pathogens

## Abstract

*Campylobacter* is the most common cause of bacterial foodborne illness in the EU, primarily linked to poultry. To better understand its transmission dynamics, we applied whole-genome sequencing (WGS) to *Campylobacter* isolates collected at slaughterhouses over a two-year period from broilers originating from 26 Danish farms. The samples included cloacal swabs and boot sock samples from broiler houses and surrounding farm environments. We identified 150 distinct cgMLST types among 883 isolates. While most cgMLST types were flock-specific, some persisted across production cycles or appeared at different farms, indicating entrenched contamination or potential common-source introductions. Notably, 39% of broiler-associated cgMLST types overlapped with human clinical isolates from the same period, with the strongest overlap among persistent and cross-farm types, particularly in conventional production systems. Our findings underscore the need for strengthened biosecurity, targeted surveillance of high-risk genotypes, and real-time WGS integration to mitigate the burden of human *Campylobacteriosis*. This study supports a One Health approach to managing zoonotic risk in poultry production.

## 1. Introduction

*Campylobacter* is the leading cause of bacterial foodborne gastroenteritis in the European Union (EU), with human infections primarily linked to contaminated poultry products, particularly chicken [1,2]. In Denmark, approximately 5000 human cases are reported annually, except for the pandemic years in 2020–2021 [2], although the true incidence is estimated to be significantly higher [3].

Of the many species within the *Campylobacter* genus, *Campylobacter jejuni* and *Campylobacter coli* are the most commonly associated with human disease. *C. jejuni* accounts for the vast majority of human gastroenteritis cases globally and in Denmark, while *C. coli* is less prevalent but still clinically relevant [1]. Both species are found in poultry, but *C. jejuni* is the dominant species in broiler chickens [4]. In animals, these species are typically asymptomatic colonizers, yet their zoonotic potential poses a significant food safety concern [1].

Since poultry is the primary reservoir of *Campylobacter*, understanding its persistence and transmission dynamics in broiler production is critical for reducing human infections. Transmission in broiler farms may occur through multiple routes, including contaminated water sources, nearby livestock, and improperly cleaned transport crates [5,6,7]. Internal barn contamination is a key factor, while environmental reservoirs outside broiler houses may contribute sporadically [5]. While on-farm interventions, such as biosecurity measures and improved hygiene practices, significantly reduce *Campylobacter* prevalence in broiler flocks [6], complete eradication remains challenging.

One of the most effective tools for studying *Campylobacter* transmission and persistence is whole-genome sequencing (WGS), which has transformed molecular epidemiology by enabling precise characterization of *Campylobacter* populations, including their genetic diversity, persistence, and transmission patterns [8,9,10,11,12]. Since 2019, Denmark has integrated WGS into the national surveillance, enabling the linkage of human *Campylobacter* infections to poultry sources, thereby strengthening outbreak detection and epidemiological investigations [10]. As part of routine surveillance, stool samples from approximately 10–15% of human *Campylobacter* cases undergo culture and WGS analysis to identify and monitor *Campylobacter* cgMLST types [10,13]. This approach has enhanced outbreak detection, transmission tracking, and intervention strategies. Further studies in European poultry slaughterhouses underscore the role of WGS in tracing contamination sources and transmission routes within broiler production [6,14,15].

Longitudinal studies suggest that *Campylobacter* strains can persist beyond individual broiler flocks. A study in a Swiss poultry slaughterhouse found that, despite high genetic diversity among isolates from broiler carcasses, a recurring *C. jejuni* lineage persisted across multiple years and farming systems, indicating that slaughterhouse environments may act as reservoirs for specific clones [16]. Similarly, persistent *C. jejuni* lineages linked to human infections have been identified in Luxembourg [17], and *Campylobacter* genotypes persisting across multiple fattening periods have been observed in German broiler farms, highlighting environmental persistence despite cleaning and disinfection efforts [5]. Previous studies have identified persistent *Campylobacter* strains in broiler farms that remain detectable across multiple rotations, suggesting the presence of stable environmental reservoirs and insufficient decontamination practices [6]. Persistence is not limited to individual farms. Cross-farm transmission has also been reported, likely facilitated by contaminated equipment, personnel movement, or shared supply chains [6]. The ability of *Campylobacter* to endure beyond single production cycles and across farms presents significant challenges for eradication. Understanding the genomic epidemiology of *Campylobacter* in broiler production is crucial for mitigating the risk of foodborne transmission, and the above findings underscore the need for genomic surveillance to monitor *Campylobacter*’s persistence and cross-farm spread.

To address these challenges, this study applied WGS to assess the prevalence and genetic diversity of *Campylobacter* in Danish broiler production over a two-year period. To achieve high-resolution genetic characterization, we applied core-genome multilocus sequence typing (cgMLST) to enable detailed phylogenetic clustering and comparison of isolates from animals, humans, and environmental sources. By integrating epidemiological data with genomic analyses, we aimed to characterize *Campylobacter*’s distribution and the persistence of cgMLST types across farms and rotations.

This study aimed to generate high-resolution genomic data on *Campylobacter* populations circulating in Danish broiler production. Such knowledge is essential to inform targeted control strategies and, ultimately, contribute to reducing the risk of *Campylobacter*-related human infections through improved surveillance and intervention planning.

## 2. Materials and Methods

### 2.1. Study Design and Sampling

This study included a total of 26 unique broiler farms, comprising 17 conventional farms and 9 free-range farms, sampled during 2020 and 2021. Of these, 15 conventional and 2 free-range farms were included in 2020. In 2021, seven new free-range farms (F4 to F10) and two new conventional farms (C15 and C17) were added, and seven farms were sampled in both years (C2, C4, C7, C8, C10, C12, and C14).

Sampling was conducted in collaboration with three Danish slaughterhouses. Farms were selected based on previous instances of *Campylobacter*, specifically targeting farms likely to test positive. Each farm had 2–6 broiler houses, each housing up to 45,000 animals. In our study, broilers were sampled at slaughter after each rotation (production cycle) during April–November in 2020. This period was chosen due to the higher incidence rates of *Campylobacter* during these months. Conventionally raised broiler flocks were typically slaughtered at 5–6 weeks of age, while free-range flocks were reared for at least 8 weeks. A visual overview of the farms and their inclusion timeline is presented in Figure 1.

At the slaughterhouse, immediately after killing, samples were collected from the cloacae using FecalSwab™ (Copan Italia S.p.A., Brescia, Italy), with each swab used on two carcasses. A total of 10 samples were collected from each flock, except for 42 partially depopulated (thinned) flocks, where 20 samples were collected (10 from each of two slaughter events).

In 2021, boot sock samples were additionally collected inside selected broiler houses 7–10 days before slaughter. These samples, collected in pairs of moistened socks by farm staff, were suspended in approximately 70 mL of modified Cary–Blair medium (SSI Diagnostica, Hillerød, Denmark) for transport. The purpose was to assess whether the cgMLST types identified at slaughter were present in the broiler houses prior to slaughter. In some cases, additional samples were collected 2–4 days before slaughter. The results from cloacal swabs and boot socks were combined to evaluate the presence of *Campylobacter* in specific flocks, as both types of samples represented findings from chicken feces of the same flock.

In some instances, boot sock samples were also collected from the external environment around the farms, following the same collection and handling procedures but performed by a slaughterhouse representative. Additional samples were collected from specific farms: pig feces at farm C14, cattle feces from a neighboring field at farm C8, and various sample types at farm C15 (details specified later).

### 2.2. Microbiological Analyses

For cultivation, all samples were transported to the laboratory at Statens Serum Institut (SSI) within 24 h. The culturing process, based on a modified version of ISO 10272-1:2017 [18], involved enrichment in selective Bolton broth (Thermo Fisher Diagnostics (Oxoid Ltd.), Basingstoke, UK) under microaerophilic conditions at 41.5 °C for 24 h, followed by plating on selective mCCDA medium (Thermo Fisher Diagnostics) incubated for 24 h under the same microaerophilic conditions at 41.5 °C. For each sample showing *Campylobacter* growth, a single colony was selected and subcultured to obtain a pure isolate. Species determination was conducted using MALDI-TOF MS (Bruker Daltonik GmbH, Bremen, Germany) prior to whole-genome sequencing (WGS). A broiler flock was considered positive for *Campylobacter* if one or more of the ten samples were culture-positive. The results from cloacal swabs and boot socks were considered jointly when determining flock status, as both sample types represent fecal material from the same birds. The presence of any of the three targeted species (*C. jejuni*, *C. coli*, or *C. lari*) qualified the flock as positive.

### 2.3. Whole-Genome Sequencing and Comparative Analysis with Human Isolates

For broiler flocks that tested positive for *Campylobacter*, three representative isolates of *C. jejuni* (if available) and one isolate of *C. coli* (if detected), as determined by MALDI-TOF, underwent whole-genome sequencing (WGS). An exception was made for the first part of 2020, during which all obtained isolates were sequenced to provide comprehensive data on *Campylobacter* strains present.

Genomic DNA was extracted from pure cultures using the DNeasy Blood & Tissue Kit (QIAGEN GmbH, Hilden, Germany). Sequencing was conducted on the Illumina NextSeq platform (Illumina, Inc., San Diego, CA, USA) using the Nextera XT Library preparation kit (Illumina), generating paired-end reads of 150 base pairs. The quality of the sequences was assessed using Bifrost, a standardized in-house pipeline (available at http://github.com/ssi-dk/bifrost, accessed on 30 June 2025), which ensured appropriate genome sizes (1.6 to 1.9 Mbp), sufficient assembly quality (fewer than 500 contigs), and contamination below 5% of reads from other genera for each sequenced isolate.

Additionally, a 7-locus multilocus sequence typing (MLST) analysis was performed for all sequences, using the MLST module in the Bifrost pipeline. In cases where no sequence type (ST) could be assigned through Bifrost, the MLST profile was determined using BioNumerics version 8.1 (bioMérieux, Applied Maths NV, Sint-Martens-Latem, Belgium). Further analysis and comparison of the isolates were conducted using BioNumerics (bioMérieux), specifically utilizing core-genome MLST (cgMLST) based on the 1343-locus Oxford scheme [19]. Genetic clusters were defined as previously described in the Danish surveillance of *Campylobacter* from patients and food [13]. This high-resolution typing framework allows for detailed phylogenetic clustering and comparison of cgMLST types across isolates from animals, humans, and environments. Throughout this manuscript, the term “type” refers to a unique cgMLST type—i.e., an allelic profile based on the Oxford 1343-locus scheme with ≤4 allelic differences—unless otherwise specified.

As part of this surveillance program, we, at SSI, routinely perform WGS on approximately 600 *Campylobacter* isolates from human cases annually. These isolates are processed using the same sequencing protocols, cgMLST scheme (the Oxford 1343-locus scheme), and analytical pipeline as described above.

To evaluate potential associations between broiler and human cases, *Campylobacter* types (cgMLST profiles) from broiler isolates were compared with those in the human surveillance dataset from the corresponding period (2020–2021). Matching was defined by identical cgMLST types (i.e., ≤4 allelic differences), and the degree of overlap was assessed in relation to both type categories and broiler production types.

Raw sequencing data for the broiler isolates are available in the NCBI Sequence Read Archive (SRA) under BioProject ID PRJNA1272010.

## 3. Results

### 3.1. Sample Collection Overview

Over the course of two years, a total of 4388 samples (4288 cloacal swabs and 100 boot sock samples) were collected, representing broilers from 26 farms. Of these, 2727 samples (62%) tested negative, while 1661 samples were positive for *Campylobacter* species (*C. jejuni*, *C. coli*, or *C. lari*). Species identification for all positive samples was performed using MALDI-TOF, as described in Section 2.2.

In 2020, species identification was performed for 896 of the 934 positive samples, identifying 88% *C. jejuni* (787/896), 11% *C. coli* (102/896), and 1% *C. lari* (7/896). In 2021, species was successfully determined for all 727 positive samples, identifying 83% *C. jejuni* (600/727), 17% *C. coli* (126/727), and <1% *C. lari* (1/727).

In 2020, a total of 2819 cloacal swab samples were collected from broilers, representing 240 flocks across 17 farms. These samples covered 3 to 5 rotations. Two of these farms were free-range (F1 and F2), while the remaining 15 farms were conventional (C1 to C14 and C16). In 2021, a total of 1569 samples were collected from broilers, representing 147 flocks across 18 farms. This included nine free-range farms, two of which had also been sampled the previous year (F1 and F2), and nine conventional farms (C2, C4, C7, C8, C10, C12, and C14), along with two new farms (C15 and C17). Of the total, 1469 samples were cloacal swabs, as in the previous year, and an additional 100 samples were collected as boot socks (in pairs) from inside the broiler houses prior to slaughter.

In the study period, 25 of the 26 farms had at least one positive flock (Figure 1). As defined in the Materials and Methods Section, a flock was considered *Campylobacter*-positive if at least one out of ten collected cloacal swab samples or one boot sock sample from inside the house was culture-positive. Furthermore, 43% (135/313) of conventional flocks and 70% (52/74) of free-range flocks tested positive for *Campylobacter* at slaughter (Figure 1). Overall, the 387 broiler flocks included in this study represented approximately 6% of the broiler flocks recorded in Denmark during 2020 and 2021 [2].

### 3.2. Campylobacter Isolate Diversity Across Farms

This section describes the overall genetic diversity of *Campylobacter* isolates collected across all farms, based on cgMLST clustering. We also introduce the categorization of cgMLST types (hereafter referred to simply as “types”) into flock, farm, persistent, and cross-farm types to enable structured interpretation of transmission patterns.

A total of 871 isolates from samples collected from animals, either at slaughter (via cloacal swabs) or within farms (via boot socks), were sequenced. The genome sequences included 756 *C. jejuni*, 111 *C. coli*, and 4 *C. lari*. Of these, 586 isolates were collected in 2020, while 285 were obtained in 2021. Additionally, 12 isolates obtained from the “additional samples” collected in 2021 were sequenced and represented 8 *C. jejuni* and 4 *C. coli* genomes.

Overall, 150 distinct types, as determined by cgMLST clusters, were identified within the dataset. Of these, 143 types were directly associated with the broiler flocks (animals and housing) across 25 of the 26 sampled farms (Figure 2). The remaining seven types were found exclusively among the additional samples. Figure 2 summarizes the distribution of the 143 cgMLST types across farms and type categories. Additional information on the individual types is provided in Appendix A.

Among the 143 detected cgMLST types, 91 (64%) were classified as “flock types,” meaning they were found in only one flock (within a single house on one farm during a single rotation). Another 21 types (15%) were categorized as “farm types,” specific to one farm within a single rotation but detected across multiple houses. Similarly, 21 types (15%) were identified as “persistent farm types,” detected on the same farm across multiple rotations, typically found in multiple houses. The remaining 10 types (7%) were classified as “cross-farm types,” observed at more than one farm, either simultaneously or at different times (Figure 2).

Out of the 143 types, 99 (69%) were found in conventional broilers, while 43 types (30%) were associated with free-range broilers. One type was identified in both conventional and free-range broilers.

Figure 3 provides an example from farm C10, illustrating how cgMLST types were distributed across broiler houses and production rotations. The figure shows both farm types (ST50-4, ST860-1, and ST872-1) occurring in multiple houses during the same rotation, and persistent farm types (ST52-1, ST1595-3, and ST583-1) reappearing across successive rotations. It also illustrates that multiple cgMLST types could be present simultaneously within a single house.

This classification of types laid the foundation for further analyses of persistence and cross-farm transmission.

### 3.3. Farm-Specific and Persistent Campylobacter Types

In this section, we explore whether certain cgMLST types were confined to specific flocks or houses, or persisted across rotations and housing units within the same farm.

The farm types and persistent farm types were detected in multiple houses or flocks within the same farm, indicating a shared *Campylobacter* presence across different houses (Figure 3).

Both farm types and persistent farm types were detected across various farms in this study; however, the majority (18/21) of the farm types were detected among conventional broilers, while most of the persistent farm types (13/21) were found among free-range broilers. An overview of the distribution of all cgMLST types across farms and type categories is shown in Figure 2, and associations with human infections are summarized in Table 1.

Most of the persistent farm types were detected across several houses; however, four persistent farm types (ST607-2, ST10025-1, ST1445-1, and ST825-1) were found exclusively in broilers from the same house but across different rotations, suggesting their persistence within that specific house.

The persistent farm types were generally detected in subsequent rotations (see example in Figure 3), with only two exceptions among the free-range broilers: ST1585-1 and ST267-3 (see Appendix A). ST1585-1 was detected in two different broiler houses, two rotations apart. It was first identified in one house in rotation 2 and then in the other house in rotation 4. ST267-3 was detected across both years, first detected in one broiler house in 2020 and reappearing in 2021 in the same broiler house and one additional house in the same rotation.

These results indicate that persistent cgMLST types were most frequently detected in free-range broilers, and their distribution varied between confinement to a single house or spread across multiple houses within a farm.

### 3.4. Cross-Farm Campylobacter Types

Here, we investigate cgMLST types that appeared in more than one farm, suggesting cross-farm contamination or a shared external source.

Ten types were identified that were not restricted to a single farm. These cross-farm types demonstrated a broader distribution, often being present in multiple houses on different farms.

Three cross-farm types (ST21-9, ST21-11, and ST257-5) were detected simultaneously at two different farms, but each occurred only once. Another type, ST22-2, was found at two separate farms nearly two months apart.

Six cross-farm types were present on at least one farm across multiple rotations. Notably, two of these types (ST7355-1 and ST122-1) were detected in both 2020 and 2021, indicating their sustained presence over time and highlighting the challenges in eradicating these types.

Each of the cross-farm types was specific to either free-range farms (four types) or conventional farms (five types), with no overlap between the two, demonstrating a nearly even distribution between the two farming systems. The only exception was ST22-2, which was detected at both a conventional farm and a free-range farm.

The broad distribution and recurrence of certain cross-farm types make them epidemiologically important, and their overlap with human cases is further discussed in Section 3.7.

### 3.5. Comparison of Campylobacter Types: Boot Socks vs. Cloacal Swabs

This section compares the cgMLST types detected in environmental samples (boot socks) 7–10 days prior to slaughter with those found in broiler samples (cloacal swabs) to evaluate the representativeness and added value of environmental sampling.

The results from cloacal swabs and boot socks were combined to comprehensively evaluate *Campylobacter* presence in specific flocks. Boot sock samples were collected inside the broiler houses of nine farms, with *Campylobacter*-positive samples obtained from five of them (C2, C7, C8, C10, and C15). In total, 18 different types were detected across all boot sock samples collected during this study.

Across these nine farms, 87 flocks were sampled with boot socks, and for 79 of the flocks, swab samples were also available, allowing for a direct comparison between the methods. Of the 79 flocks sampled with both boot socks and cloacal swabs, 43 flocks tested negative for *Campylobacter* in both methods, 19 flocks were positive in both, and 17 flocks were negative in boot sock samples but positive in cloacal swabs. No flocks were positive for *Campylobacter* only in boot socks.

Among the 19 flocks that tested positive for *Campylobacter* in both boot sock and cloacal swab samples, 12 flocks showed complete agreement in type, with a single, identical type detected by both methods. In four additional flocks, the same type was found in both samples, but cloacal swabs also revealed one additional type not detected in the corresponding boot sock. In the remaining three flocks, different types were identified in the two sample types. Of these, two flocks had a single, non-overlapping type in each method, while in the third flock, cloacal swabs revealed two distinct types, neither of which was detected in the boot sock sample. Among the 19 flocks that were positive for *Campylobacter* in both boot sock and cloacal swab samples, a higher number of types was observed in swabs than in boot socks. Several types were detected only in cloacal swabs and not in the corresponding boot sock samples.

This indicates a partial correspondence, where boot sock sampling mirrored some, but not all, types found in cloacal swabs from the same flocks. For example, ST7355-1 was detected in boot socks from three flocks: first, at farm C10, where it was also found in cloacal swabs from the same flock, and later, in two successive flocks at farm C7 (house 2). However, at farm C7, ST7355-1 was only detected in the cloacal swabs from the first of these two flocks, indicating that a type can be present in the environment without being detected in the broilers at slaughter.

Additionally, five types were exclusively detected by boot sock sampling and were not observed in cloacal swabs or any other samples throughout this study. This indicates that boot sock sampling provided unique insights into the environmental reservoir that cloacal swabs did not capture.

One type, ST122-1, was found in a single boot sock at farm C15 in 2021 but was absent from cloacal swabs at slaughter from this farm. Notably, ST122-1 was detected across several houses at another farm, C12, during a 2020 rotation, suggesting its persistence across different environments.

These observations show that boot sock sampling of broiler houses 7–10 days before slaughter can capture environmental cgMLST types that may not be detected in broilers at slaughter, highlighting its value for early detection and surveillance.

### 3.6. Environmental Detection of Campylobacter and Additional Sampling Results

Here, we report the findings from samples collected outside broiler houses, including manure, flies, and other environmental sources, to assess whether external reservoirs contribute to *Campylobacter* presence on farms.

In 2021, additional sampling was conducted in the environment around selected farms (C2, C7, C8, C10, C14, C15, and C17) using boot socks. Out of 98 boot sock samples collected, only 2 tested positive for *Campylobacter*, indicating a low detection rate in the external environment. At farm C15, 35 additional samples were taken from various locations, including floor cracks (10 samples), flies (10), water systems (4), heat exchangers (4), wastewater (5), mud (1), and a rain gutter (1). All tested negative for *Campylobacter*.

However, additional sampling of cattle and pig manure revealed *Campylobacter* presence. Of ten cattle manure samples from a neighboring farm (C8), six tested positive, while two pig feces samples from farm C14 were also positive. These findings suggest that cattle manure and pig feces may act as potential environmental reservoirs or sources of *Campylobacter* contamination, contributing to its spread into broiler flocks.

In total, eight distinct cgMLST types were identified from environmental samples collected outside the broiler houses across the farms. Seven of these types were unique to the external environment and were not found in any samples taken from inside the houses. The eighth type, detected in cattle manure, was the cross-farm type, ST7355-1, which had been previously identified at multiple farms in both 2020 and 2021. Of the seven unique types, three were detected from boot swab samples outside the houses at farms C7 and C14. Two types were found in a pig house associated with farm C14, while the final two were detected in cattle manure samples from farm C8.

These findings complete the overview of environmental *Campylobacter* detections outside broiler houses, highlighting that most cgMLST types identified in the external environment were not detected in animals or housing samples.

### 3.7. Overlap Between Campylobacter Types and Human Infections (2020–2021)

In the final section, we examine the overlap between cgMLST types detected in broilers and those observed in the concurrent Danish human surveillance data. This comparison allows the assessment of potential public health relevance.

In 2020 and 2021, Denmark recorded 3748 and 3734 human *Campylobacter* cases, respectively, with approximately 600 isolate sequences (*C. jejuni* and *C. coli*) generated each year. Of the 150 cgMLST types identified in this study, 58 (39%) were also detected among the human surveillance sequences from the same period.

Figure 4 illustrates the proportion of the 150 cgMLST types that overlap with those found in the human surveillance dataset, distributed across the five type categories.

A high proportion of cross-farm cgMLST types (80%) were found among concurrent patient cases, as were 62% of farm cgMLST types and 48% of persistent farm cgMLST types. Among flock types, 30% matched the human cgMLST types, while none of the additional sample cgMLST types matched human cases.

Among the 58 cgMLST types also detected in human surveillance, the majority (45 cgMLST types (78%)) were found in conventional broilers, while 12 cgMLST types (21%) were present in free-range broilers (Table 1). One cgMLST type, as previously mentioned, was found in both conventional and free-range broilers. Thus, 45% (45/99) of the cgMLST types detected in conventional broilers and 28% (12/43) of those in free-range broilers were observed in human *Campylobacter* surveillance.

Although most of the persistent types observed in this study were found among free-range broilers, only 31% (4/13) of free-range persistent types matched the human surveillance dataset, while 75% (6/8) of the persistent types observed in conventional production had human matches.

Similarly, 50% (2/4) of the free-range cross-farm types were seen in the human surveillance dataset, while 100% (5/5) of the cross-farm types in conventional production were seen among human cases.

Most types overlapping with the human side were only detected as small clusters, i.e., with a few patients affected; however, six types were represented in many human cases and considered as ongoing outbreaks at the time. Of these six, four were among the cross-farm category (ST7355-1, ST122-1, ST22-2, and ST21-9), one was a farm type (ST50-5), and the last was a flock type (ST19-4). Five of these types were detected among the conventional broilers, and the last was the one detected in both conventional/free-range types.

The high proportion of overlapping cgMLST types—particularly among cross-farm and persistent types—highlights their zoonotic potential and underscores the importance of continued genomic surveillance for public health.

## 4. Discussion

This study aimed to evaluate the diversity, persistence, and transmission of *Campylobacter* in Danish broiler farms over two years using WGS. Here, we contextualize these findings within the broader framework of control strategies and public health implications.

Effective mitigation of *Campylobacter* contamination and associated human illness depends on a clear understanding of the pathogen’s prevalence, persistence, and transmission dynamics within broiler production systems [4]. This study provides a detailed snapshot of *Campylobacter*’s diversity, persistence, and possible transmission in a selection of Danish broiler farms over a two-year period. By applying WGS to isolates from broilers, their housing environments, and selected external sources, we characterized the dynamics of *Campylobacter* populations across rotations and farming systems. WGS enabled the detection of both sporadic and persistent cgMLST types (hereafter referred to as “types”), as well as evidence of cross-farm transmission or a common-source of introduction. These findings contribute to a better understanding of *Campylobacter*’s epidemiology in poultry production and underscore the importance of investigating contamination sources, transmission pathways, and environmental or operational factors that may support bacterial persistence and spread.

### 4.1. Strain Diversity and Persistence

The predominance of unique cgMLST types that appear and disappear across individual broiler flocks highlights a highly dynamic *Campylobacter* type distribution (cgMLST types) within farms. Such transient cgMLST types, which rarely persist across rotations, likely reflect ongoing introductions from environmental or external sources [7,9,20]. This aligns with findings from *porA*-targeted sequencing studies, where highly diverse and transient *Campylobacter* populations have been observed even within single flocks, particularly among breeder chickens [21]. The origins of these sporadic types remain unclear, but their fleeting presence suggests a continual influx of novel strains rather than long-term persistence.

Despite this high turnover, our study also identified a subset of persistent cgMLST types that recurred across multiple rotations within the same farm. These findings imply the existence of entrenched contamination sources, potentially sustained by insufficient cleaning protocols, ineffective disinfection, or environmental reservoirs within broiler houses, factors previously implicated in *Campylobacter* survival [5,20]. Interestingly, some cgMLST types were detected simultaneously in multiple broiler houses within a single farm, suggesting intra-farm transmission routes. Shared water systems, biofilm-prone equipment, and personnel movement between houses have been proposed in earlier studies [5,6].

Moreover, culture-independent *porA* sequencing has demonstrated that *Campylobacter* DNA can be detected even in culture-negative flocks, suggesting that early-stage or low-abundance genotypes may be more widespread than conventional methods suggest [22]. This hidden diversity may partially explain the reappearance of certain strains and underscores the limitations of relying solely on culture-based detection in surveillance.

While persistent cgMLST types raise concern for long-term contamination, their zoonotic implications varied by production type. In our study, 75% of persistent cgMLST types in conventional broiler flocks were also found in human clinical cases, compared with only 31% in free-range systems. This suggests that persistent strains circulating in conventional production may represent a higher zoonotic risk. These findings underscore the need for targeted hygiene interventions and improved biosecurity in conventional systems to disrupt cycles of recontamination. Importantly, monitoring persistence at the farm level—beyond single-flock snapshots—is critical for identifying high-risk environments and informing more effective, long-term control strategies.

### 4.2. Zoonotic Overlap and Transmission Pathways

Of the 150 cgMLST types identified in this study, 39% overlapped with those detected in the national human surveillance program, confirming a substantial zoonotic connection between broiler production and human infections [23]. However, the true extent of overlap was likely underestimated due to limited sample coverage: our study encompassed only ~6% of Danish broiler flocks, and the national WGS surveillance program includes just ~10% of clinical cases. Nevertheless, this substantial genetic overlap reinforces the central role of poultry as a reservoir for *Campylobacter* infections in humans.

This pattern was even more pronounced in conventional broiler production, where 45% of detected cgMLST types matched human isolates, compared with 28% in free-range systems. This disparity may reflect differences in market share and consumer exposure, with conventional poultry accounting for a larger proportion of retail meat. However, it may also indicate genuine differences in transmission risk, possibly due to variations in hygiene standards, stocking densities, or environmental exposure pathways between the two systems.

Cross-farm cgMLST types—those detected on more than one farm—appeared especially concerning from a public health perspective. Eighty percent of cross-farm cgMLST types were also found in human cases, suggesting that these strains are more likely to persist, spread, and ultimately reach consumers. In several instances, these cross-farm types were linked to known outbreaks, as identified through national surveillance data and outbreak investigations conducted by Statens Serum Institut, with some of these also partially reported in the Danish Annual Report on Zoonoses [24,25]. Of the six outbreak-associated genotypes detected in our study, four were also cross-farm types, reinforcing the need to prioritize these strains in surveillance programs.

The identification of cross-farm cgMLST types also raises questions about how transmission occurs between geographically and operationally distinct broiler farms. Known vehicles of spread—such as shared transport crates, contaminated equipment, and personnel movement—have previously been implicated in inter-farm transmission [6]. A major challenge in preventing such transmission is the existence of multiple potential reservoirs and routes—both within farms and between producers—that vary depending on management practices, housing systems, and biosecurity protocols [6]. This structural variability complicates control efforts, as overlapping pathways may facilitate both the persistence and spread of high-risk genotypes.

The predominance of these strains in conventional broiler production—where operations typically include many large flocks and multiple broiler houses—highlights the urgent need for enhanced biosecurity and control measures in this sector, where cross-farm transmission is more likely to result in human illness. Further research is needed to elucidate the relative contribution of different transmission routes, including upstream sources such as hatcheries or parent flocks, and to identify critical control points for intervention.

WGS-based studies in Denmark have found similar patterns in the opposite direction—where 25–31% of clinical isolates matched concurrent cgMLST types from broilers or chicken meat [10,13]. Together, these results reinforce the substantial role of poultry in the national burden of *Campylobacteriosis*.

While our approach differs from formal source attribution models, it provides strong genomic and epidemiological evidence that *Campylobacter* strains circulating in broiler production—particularly those that are persistent or cross-farm types—contribute significantly to the human disease burden. Together, these findings underscore the importance of proactive surveillance for persistent and cross-farm strains, which appear most likely to contribute to human infection and broader spread. Prioritizing these high-risk strains in control programs may help reduce the risk of widespread transmission and human exposure.

In addition to the zoonotic risks outlined here, the role of environmental reservoirs and practical sampling strategies also warrants attention.

### 4.3. Environmental Monitoring and Early Detection

Environmental sampling played a key role in detecting *Campylobacter* cgMLST types that were not identified in broiler samples.

Although boot sock sampling did not capture the full diversity of cgMLST types found in cloacal swabs, it consistently detected the dominant genotype in most flocks that were positive in both methods. Differences between the two sampling approaches may reflect both variation in sampling timing and variation in the number of isolates sequenced from each sample type. In some cases, cgMLST types were detected in boot socks but not in cloacal swabs, indicating that environmental contamination may precede or persist beyond detectable colonization in broilers. For example, the cgMLST type ST7355-1 was identified in boot socks without being present in cloacal swabs from the same flock, pointing to environmental reservoirs as potential sources of delayed or intermittent recontamination of flocks. Together, these findings highlight that environmental detection can serve as an early indicator of risk and support the utility of boot socks as a practical early-warning tool for detecting *Campylobacter* in the broiler house environment [26].

Sampling outside broiler houses resulted in a low detection rate, but this likely reflects limitations in sensitivity or sample coverage rather than a true absence of *Campylobacter* [27]. Notably, in our study, *Campylobacter* was identified in cattle and pig manure from nearby farms, indicating potential interspecies transmission or cross-farm contamination. This observation aligns with recent Danish WGS-based studies, where cattle were highlighted as a plausible reservoir for *Campylobacter* in poultry production, while pigs appeared to play a more limited role [15]. Although the exact role of non-poultry reservoirs remains unclear, their presence supports the need for broader environmental surveillance [4].

Collectively, these findings indicate that incorporating environmental sampling—both within and outside broiler houses—can enhance surveillance sensitivity and provide earlier signals of contamination. This is particularly relevant on farms where persistent or high-risk cgMLST types have been detected, as environmental reservoirs may facilitate continued transmission. Integrating such strategies into monitoring programs could help prevent flock colonization and reduce the likelihood of broader spread [28].

### 4.4. Implications, Limitations and Future Directions

This study offers important insights for the design of *Campylobacter* surveillance and control strategies within a One Health framework.

The detection of persistent and cross-farm genotypes—many of which overlapped with human clinical cases—underscores the need to monitor contamination patterns not just at the level of individual flocks but also longitudinally at the farm level. Targeting farms with recurring or high-risk cgMLST types for intensified monitoring may help break cycles of recontamination. Likewise, the frequent detection of outbreak-associated and zoonotic cgMLST types across multiple sites points to the need for coordinated surveillance across production networks.

Environmental sampling, including boot socks and testing of nearby non-poultry sources, emerged as a valuable supplement to conventional broiler sampling. These methods proved useful in identifying cgMLST types that were not detected in cloacal swabs, offering early warning of environmental contamination. Integrating such tools into standard surveillance protocols could enhance detection sensitivity and help identify farms at risk before full flock colonization occurs.

Despite these strengths, several limitations should be considered when interpreting our findings. This study encompassed only ~6% of Danish broiler flocks during the study period and ~10% of clinical human cases nationally, limiting the ability to capture all relevant transmission events. Free-range flocks were underrepresented, which restricts the generalizability of comparisons between production systems. Additionally, while cloacal swabs and boot socks provided valuable isolate collections, reliance on culture-based methods likely underestimated total *Campylobacter* diversity, as strains that failed to grow under laboratory conditions were excluded.

WGS enabled high-resolution genotyping of isolates, but this study did not include functional genomic analyses, such as antimicrobial resistance (AMR) or virulence profiling. Future work should explore these aspects using the deposited sequence data to better understand the clinical relevance and risk profile of circulating genotypes. Broader implementation of culture-independent approaches, including metagenomic sequencing, could also reveal hidden diversity and offer new tools for surveillance.

Together, these findings support a shift toward more integrated, longitudinal, and genomically informed surveillance strategies—both within and across farms—that prioritize high-risk strains and adapt to the dynamic nature of *Campylobacter* epidemiology in poultry production.

## 5. Conclusions

The use of WGS in this study has provided valuable insights into the diversity, spread, and persistence of *Campylobacter* in Danish broiler production. The strong overlap between broiler-associated genotypes and those found in human infections highlights the need for a coordinated One Health approach to surveillance and control. Conventional production systems, in particular, were associated with a higher prevalence of human-linked strains, including those linked to outbreaks and cross-farm transmission events. To reduce the burden of human *Campylobacteriosis*, stricter biosecurity protocols are needed, especially in conventional production, where the risk of persistence and inter-farm spread appears greatest. Surveillance systems should prioritize high-risk genotypes—particularly persistent and cross-farm strains—regardless of their current prevalence. However, before more effective management strategies can be implemented, clarification of transmission routes is essential to ensure targeted and evidence-based interventions.

Finally, even small or intermittent contamination events should not be overlooked, as low-prevalence strains can still lead to human infections and outbreaks under the right conditions. To strengthen surveillance, future efforts should incorporate targeted genotyping of high-risk genotypes and explore the integration of real-time WGS in broiler production. Such systems could enable early detection of persistent and zoonotic strains and support more agile, evidence-based interventions.

## Figures and Tables

**Figure 1 pathogens-14-00821-f001:**
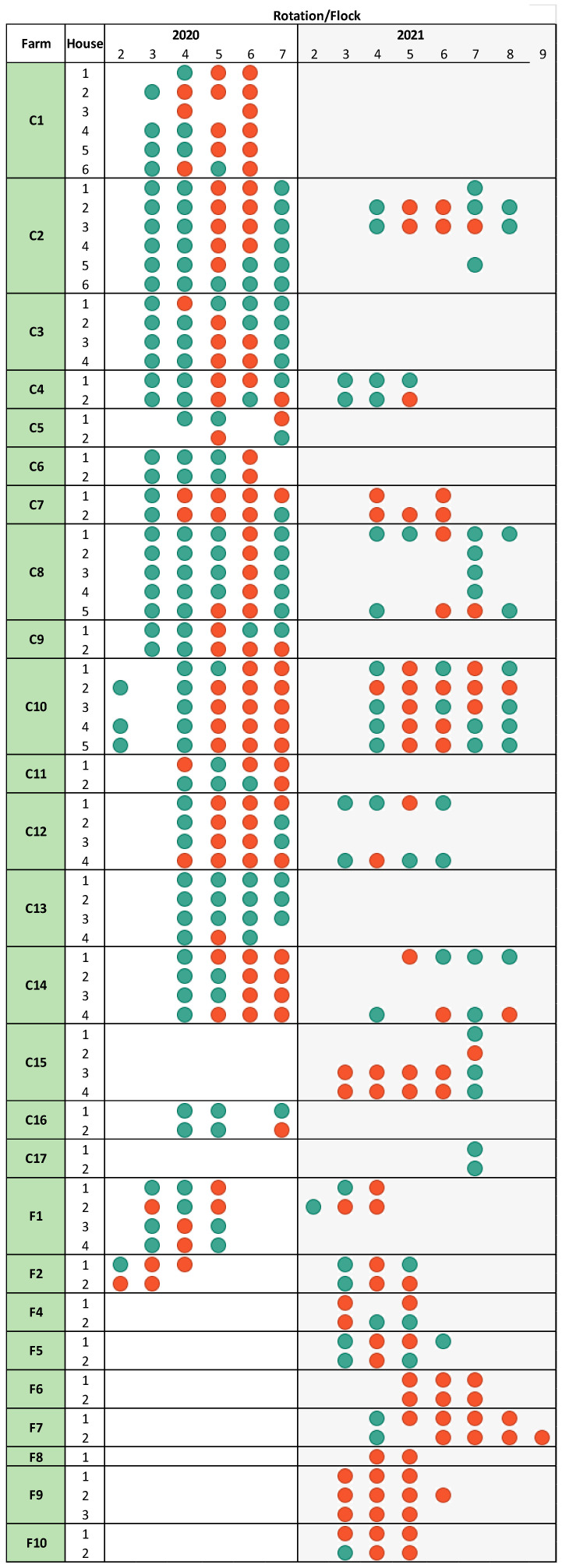
*Campylobacter* status of broiler flocks sampled in Denmark (2020–2021), by farm, house, and production rotation. Each circle represents a single flock from a specific broiler house and production rotation. Red circles indicate *Campylobacter*-positive flocks; green circles indicate *Campylobacter*-negative flocks. Numbers 2–7 in 2020 and 2–9 in 2021 (top bar) indicate rotation numbers. One positive sample from farm C1 (house 6; rotation 4) was lost and not sequenced.

**Figure 2 pathogens-14-00821-f002:**
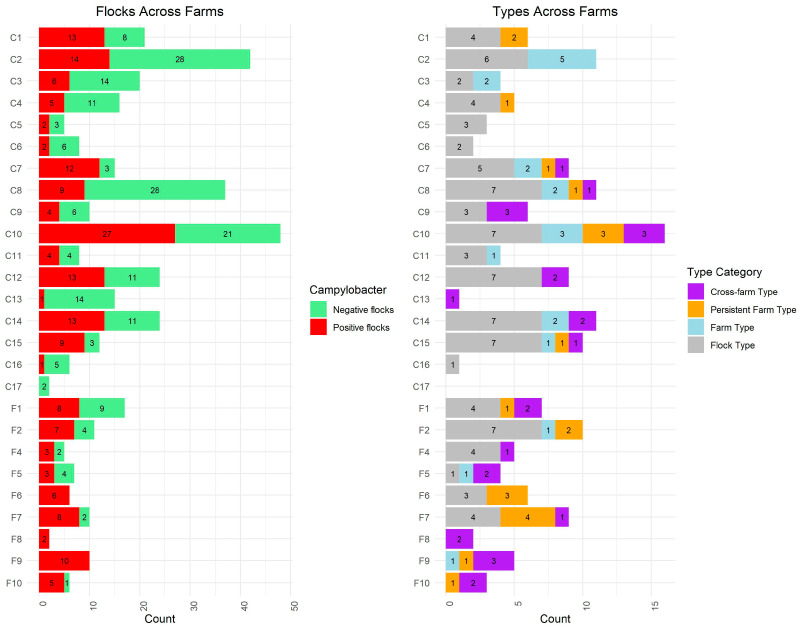
*Campylobacter* status and distribution of detected cgMLST types across Danish broiler farms (2020–2021). The first panel shows the proportion of *Campylobacter*-positive and -negative flocks for each farm. The second panel shows the number of cgMLST types per farm, categorized into epidemiological type categories: cross-farm type, persistent farm type, farm type, or flock type.

**Figure 3 pathogens-14-00821-f003:**
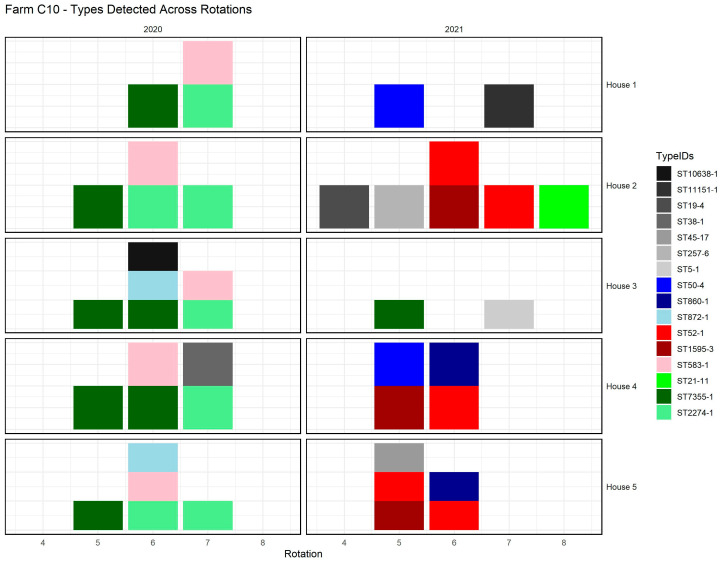
Distribution of detected cgMLST types across rotations and broiler houses at farm C10 (2020–2021). Each color represents a specific cgMLST type. Grey indicates flock types detected only once. Other colors denote broader epidemiological categories: farm types (blue), persistent farm types (red), and cross-farm types (green).

**Figure 4 pathogens-14-00821-f004:**
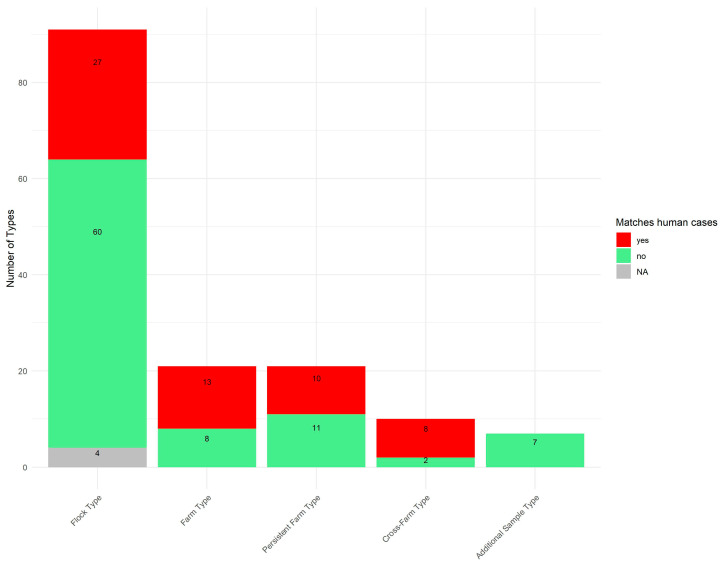
Proportion of study cgMLST types detected in concurrent Danish human *Campylobacter* surveillance (2020–2021). The 150 cgMLST types identified in broiler, environmental, and additional samples during this study are shown, grouped by the five defined type categories (cross-farm type, persistent farm type, farm type, flock type, and additional sample type). Red bars indicate cgMLST types also detected in human surveillance isolates from the same period, while green bars indicate cgMLST types with no human matches. NA denotes *C. lari* cgMLST types, which were not included in the comparison.

**Table 1 pathogens-14-00821-t001:** Distribution of *Campylobacter* cgMLST types by type category, production system, and their association with human infections (Denmark, 2020–2021). The table shows cgMLST types grouped into four epidemiological categories: flock, farm, persistent farm, and cross-farm types. Counts are presented for conventional (C), free-range (F), and types detected in both (C/F) production systems. “Types matching human cases” indicates cgMLST types also found in the concurrent Danish human *Campylobacter* surveillance. “Flocks per type” is the observed range of flocks in which each cgMLST type occurred. “Total human cases” and “human cases per type” refer to the number of matching human isolates during the study period.

Type Category	Category Specifics	Conventional (C)	Free-Range (F)	C/F
Flock types (no. = 91)	Types in category	68	23	-
	Types matching human cases	22	5	-
	Flocks per type	1	1	-
	Total human cases	56	11	-
	Human cases per. type	0.8	0.5	-
Farm types (no. = 21)	Types in category	18	3	-
	Types matching human cases	12	1	-
	Flocks per type	2–5	2	-
	Total human cases	39	1	-
	Human cases per. type	2	0.3	-
Persistent farm types (no. = 21)	Types in category	8	13	-
	Types matching human cases	6	4	-
	Flocks per type	2–5	2–8	-
	Total human cases	40	12	-
	Human cases per. type	5	0.9	-
Cross-farm types (no. = 10)	Types in category	5	4	1
	Types matching human cases	5	2	1
	Flocks per type	2–11	3–9	6
	Total human cases	82	6	12
	Human cases per. type	16	1.5	12

## Data Availability

The sequence data generated in this study are available in the NCBI Sequence Read Archive (SRA) under BioProject ID PRJNA1272010.

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
