# Peer review of "The Clonal Spread and Persistence of Campylobacter in Danish Broiler Farms and Its Association with Human Infections"

_pathogens, 2025, doi:10.3390/pathogens14080821_

Round 1

Reviewer 1 Report

Comments and Suggestions for Authors

This is an interesting, comprehensive, and well-written article about novel Campylobacter research. WGS provides new ways of investigating bacterial transmission routes and compare broiler samples with human samples. 

The text is easy to read. The figure captions can be improved to explain better the figures, i.e. figure 1, right colomn showing year 2020 and 2-7 and 2021 and 2-9. What do 2-7 and 2-9 mean? Add in caption. 

Maybe express even clearer: how do you interpret your major findings? Is 39 % overlap of genotypes with human isolates high or low or as expected? Which recommondations do you send to the broiler meat producers after this study?

Author Response

Comments 1: This is an interesting, comprehensive, and well-written article about novel Campylobacter research. WGS provides new ways of investigating bacterial transmission routes and compare broiler samples with human samples. 

Response 1: We greatly appreciate the reviewer’s positive overall assessment and constructive suggestions. We have addressed all points in detail below, improving figure clarity, clarifying key interpretations, and expanding the discussion to better convey the practical implications of our findings. We believe these revisions fully address all concerns and further strengthen both the clarity and practical relevance of the manuscript. Detailed responses to each point are provided below.

Comments 2: The text is easy to read. The figure captions can be improved to explain better the figures, i.e. figure 1, right colomn showing year 2020 and 2-7 and 2021 and 2-9. What do 2-7 and 2-9 mean? Add in caption. 
Response 2: Thank you for the comment. We completely agree and have adapted the captions of all figures.

Comments 3: Maybe express even clearer: how do you interpret your major findings? Is 39 % overlap of genotypes with human isolates high or low or as expected?

Response 3: We thank the reviewer for this question. In Section 4.2, we now clarify that the 39 % overlap should be considered substantial and is likely underestimated due to limited sampling (~6 % of broiler flocks and ~10 % of human cases). While earlier Danish WGS studies reported 25–31 % overlap when comparing in the opposite direction (human-to-broiler match), our higher figure—based on broiler-to-human match—further reinforces the central role of poultry as a reservoir for human Campylobacter infection in Denmark.

Comments 4: Which recommondations do you send to the broiler meat producers after this study?

Response 4: We thank the reviewer for this question. Practical recommendations for broiler meat producers have been incorporated into the revised discussion (Sections 4.2–4.4). Based on our findings, we propose the following strategies to strengthen prevention and control of Campylobacter on broiler farms:

  • Longitudinal farm-level monitoring – Track contamination patterns across multiple flock rotations to identify and target high-risk farms.
  • Prioritization of high-risk genotypes – Intensify monitoring and control efforts for persistent or cross-farm cgMLST types, particularly those detected in human cases or linked to outbreaks.
  • Integration of environmental sampling – Apply sampling both inside broiler houses and in surrounding environments (including non-poultry reservoirs) as early-warning tools to detect contamination before flock colonisation.
  • Coordinated surveillance across the production network – Strengthen information sharing and coordinated monitoring between farms, slaughterhouses, and public health surveillance systems.
  • Broader genomic and rapid-response approaches – Incorporate culture-independent methods to detect high-risk strains earlier and enable interventions before full flock colonization, breaking potential transmission chains.

These recommendations are grounded in our WGS-based findings and are intended to support broiler producers in reducing the persistence and spread of high-risk Campylobacter genotypes.

Reviewer 2 Report

Comments and Suggestions for Authors

The manuscript submitted by Joensen et al. aimed to describe the clonal dissemination and persistence of Campylobacter in broiler farms and its association with human infections.

The topic is highly relevant, as Campylobacter is one of the leading etiological agents of foodborne illness globally, also causing significant economic impacts related to control measures in poultry farms.

The study is strengthened by an appropriate sampling strategy (26 farms), a large number of isolates, and the use of whole genome sequencing (WGS), the most robust method available for the study objectives.

However, the manuscript is poorly organized and difficult to follow, particularly in its current structure and presentation. While the study clearly involved substantial effort and generated a large volume of data, the authors appear to struggle in extracting and organizing the key findings to construct a coherent and impactful discussion. Moreover, the interpretation of results and their comparison with existing literature are limited and fragmented.

Major Remarks

  • Relationship with human cases: My major concern is that the authors claim to analyze the association with human infections; however, this appears to be more of a discussion rather than an objective supported by the methodology. There is no information on the origin or sequencing of human isolates in the Methods section. The authors must clarify whether such data were obtained from external sources and detail the methodology used for comparative analyses. Alternatively, they should remove this objective from the stated aims (and others sections) of the study and address it only in the discussion, if applicable.
  • Use of WGS: It is unclear why the authors performed whole genome sequencing (WGS) solely to determine sequence types (STs). WGS provides a comprehensive dataset that allows for deeper analyses, including phylogenetic reconstruction and the investigation of genetic markers associated with virulence and antimicrobial resistance. The manuscript would greatly benefit from such analyses, which are essential to fully explore the epidemiological significance of the isolates. The authors should clarify their rationale for limiting the scope of WGS analyses and are strongly encouraged to include additional genomic investigations if the data are available.
  • Manuscript structure: Despite the robustness of the study design and data, the manuscript is structured in a confusing manner—particularly the Results section, which is difficult to follow and interpret.
  • Discussion section: The discussion is fragmented into several subsections, each addressing limited aspects with insufficient depth. I recommend reorganizing this section into a single, cohesive narrative, allowing for more comprehensive exploration of the findings and better integration with the literature.
  • Recommendations for control: The authors should expand the discussion to include practical implications for Campylobacter control. Based on the findings, what strategies can be proposed to improve prevention and control measures on broiler farms? How can this data inform surveillance or biosecurity practices?
  • Ethical approval: The manuscript does not include the name of the ethics committee nor the corresponding approval number. This information is essential for studies involving animal sampling and must be clearly stated in accordance with the journal’s requirements and international ethical standards. The authors should include these details in a dedicated section at the end of the Materials and Methods, as specified in the journal’s guidelines.

Minor Remarks

  • Line 21: Remove italics from the term campylobacteriosis.
  • Keywords: Eliminate any terms already present in the title and ensure the number of keywords adheres to journal guidelines.
  • Lines 53–55: This paragraph is too short and should be merged with adjacent text.
  • Lines 74–84: These lines should be consolidated into a single paragraph for better flow.
  • Introduction: It is important to clarify two key points: (i) the main Campylobacter species involved in foodborne illnesses, and (ii) that poultry commonly serve as asymptomatic carriers of this pathogen.
  • Lines 91–93: The relevance of this information is unclear; please clarify or remove.
  • Line 87: Verify numerical consistency—26 farms are mentioned in the abstract, but this is not clearly reflected in the main text. Although this number appears in Figure 1, the sampling design across farms and slaughterhouses must be explained more clearly in the Methods section.
  • Section 2.2: Rename this section to “Microbiological Analyses.” The isolation method used should be properly cited.
  • Line 134: Provide details on the presumptive differences used for species characterization.
  • Section 2.4: This section should be relocated to the end of the manuscript as per the journal’s submission guidelines. Include the name of the ethics committee and approval number.
  • Throughout the manuscript and figures, replace generic references to “type” with the correct notation for sequence types (ST).
  • Line 402: Reconsider the use of the term “vectors”, which typically refers to invertebrate organisms; use more appropriate terminology if referring to vehicles of transmission.
  • Line 414: Provide references for the outbreaks mentioned.
  • Section 4.5: Include proper references to support the content presented.
  • Improve all table titles to ensure they are self-explanatory and understandable without reference to the main text.

Author Response

Comments 1: The manuscript submitted by Joensen et al. aimed to describe the clonal dissemination and persistence of Campylobacter in broiler farms and its association with human infections. The topic is highly relevant, as Campylobacter is one of the leading etiological agents of foodborne illness globally, also causing significant economic impacts related to control measures in poultry farms. The study is strengthened by an appropriate sampling strategy (26 farms), a large number of isolates, and the use of whole genome sequencing (WGS), the most robust method available for the study objectives. However, the manuscript is poorly organized and difficult to follow, particularly in its current structure and presentation. While the study clearly involved substantial effort and generated a large volume of data, the authors appear to struggle in extracting and organizing the key findings to construct a coherent and impactful discussion. Moreover, the interpretation of results and their comparison with existing literature are limited and fragmented.

Response 1: We thank the reviewer for recognizing the relevance of our study, the robustness of the sampling design, and the value of using WGS for our objectives. We also appreciate the constructive feedback on improving the manuscript’s organization, clarity, and integration with the literature. In the revisions, we have restructured the Results and Discussion for better readability, expanded methodological descriptions, clarified the human isolate comparisons, and strengthened the interpretation and contextualization of our findings. We believe these extensive revisions have addressed all concerns, clarified methodology and structure, and strengthened the manuscript’s clarity, interpretive depth, and public health relevance. Detailed responses to each point are provided below.

Major Remarks

Comments 2: Relationship with human cases: My major concern is that the authors claim to analyze the association with human infections; however, this appears to be more of a discussion rather than an objective supported by the methodology. There is no information on the origin or sequencing of human isolates in the Methods section. The authors must clarify whether such data were obtained from external sources and detail the methodology used for comparative analyses. Alternatively, they should remove this objective from the stated aims (and others sections) of the study and address it only in the discussion, if applicable.

Response 2: We appreciate this important and constructive comment. We agree that the original Methods section did not sufficiently describe the origin and handling of the human isolate data used for comparison. To clarify, all human isolates referenced in the comparative analyses were sequenced in-house as part of the Danish national Campylobacter surveillance program at Statens Serum Institut. Approximately 600 clinical isolates are sequenced annually, and the data undergo the same sequencing and analytical pipeline (including cgMLST) as the broiler isolates in the current study.
To address this, we have now expanded Section 2.3 of the Methods to explicitly describe the origin of the human data, the sequencing procedures, and the methodology used to compare broiler and human isolates. Specifically, we have included the following clarifications:

  • The human sequences were generated through our routine national surveillance.
  • The same cgMLST pipeline and scheme (Oxford 1,343-locus) were applied.
  • Matching between broiler and human isolates was based on identical (≤ 4AD) cgMLST profiles.
  • The comparison focused on assessing overlap in type distribution across defined type categories and broiler production types.

We have also slightly revised the subsection title to reflect the inclusion of this analysis (Whole-Genome Sequencing and Comparative Analysis with Human Isolates). We believe these additions now provide a clear methodological basis for the analyses and conclusions presented in the Results and Discussion.

Comments 3: Use of WGS: It is unclear why the authors performed whole genome sequencing (WGS) solely to determine sequence types (STs). WGS provides a comprehensive dataset that allows for deeper analyses, including phylogenetic reconstruction and the investigation of genetic markers associated with virulence and antimicrobial resistance. The manuscript would greatly benefit from such analyses, which are essential to fully explore the epidemiological significance of the isolates. The authors should clarify their rationale for limiting the scope of WGS analyses and are strongly encouraged to include additional genomic investigations if the data are available.

Response 3: We thank the reviewer for raising this important point regarding the potential applications of WGS. We agree that WGS enables a broad range of downstream analyses beyond classical 7-locus MLST. We would like to emphasize that our genomic analysis was not limited to classical MLST. Instead, we employed core-genome multilocus sequence typing (cgMLST) based on the 1,343-locus Oxford scheme, enabling high-resolution phylogenetic clustering and comparison of Campylobacter isolates from broilers, humans, and the environment. This approach formed the basis for key analyses in the manuscript, including the identification of persistent genotypes, detection of cross-farm transmission, and comparison with concurrent human cases.

To clarify this more explicitly in the manuscript, we have made the following additions:

  • In the Introduction, we added the sentence:

“To achieve high-resolution genetic characterization, we applied core-genome multilocus sequence typing (cgMLST) to enable detailed phylogenetic clustering and comparison of isolates from animals, humans, and environmental sources.”

  • In the Methods section (2.3), we inserted:

“This high-resolution typing framework allows for detailed phylogenetic clustering and comparison of cgMLST types across isolates from animals, humans, and environments.”

  • In the Limitations section, we added:

“In addition, while our WGS analyses focused on cgMLST-based high-resolution typing to investigate persistence and zoonotic overlap, we did not include functional genomic analyses (e.g. antimicrobial resistance or virulence profiling). These aspects lie beyond the scope of this study but may be explored in future work using the deposited sequence data.”

We believe these additions clarify that the WGS data were fully leveraged for high-resolution analyses, and that further genomic investigations can be pursued using the publicly available sequences (BioProject PRJNA1272010).

Comments 4: Manuscript structure: Despite the robustness of the study design and data, the manuscript is structured in a confusing manner—particularly the Results section, which is difficult to follow and interpret.

Response 4:  We thank the reviewer for this helpful comment and fully agree that a well-structured Results section is essential for interpretation and reader engagement. To improve clarity and navigation, we have revised the Results section by:
• Adding short introductory sentences at the beginning of each subsection (3.2–3.7) to clarify the purpose and focus of the analysis.
• Including brief interpretive summary statements at the end of relevant subsections to enhance readability and guide the logical flow.
• Revising figure references to better align with the surrounding text. While no structural changes were made to the figures themselves, we fine-tuned transitions and figure callouts—particularly in Sections 3.2 and 3.3—to ensure that each figure is introduced in a clear and contextually appropriate manner.

Although we considered consolidating subsections 3.2–3.4, we ultimately decided to retain the current structure to maintain clarity in the stepwise analytical progression (diversity → persistence → cross-farm spread).

Specifically, the following new sentences have been added:

  • Section 3.2 (Campylobacter Isolate Diversity Across Farms):
  • At the beginning of the subsection: “This section describes the overall genetic diversity of Campylobacter isolates collected across all farms, based on cgMLST clustering. We also introduce the categorization of genotypes into flock, farm, persistent, and cross-farm types to enable structured interpretation of transmission patterns.

  • At the end of the subsection: “This classification of types laid the foundation for further analyses of persistence and cross-farm transmission.”

Section 3.3 (Farm-Specific and Persistent Campylobacter Types):

  • At the beginning of the subsection:” In this section, we explore whether certain Campylobacter types were confined to specific flocks or houses, or persisted across rotations and housing units within the same farm.”
  • At the end of the subsection:These results indicate that persistent genotypes were most frequently detected in free-range broilers, and their distribution varied between confinement to a single house or spread across multiple houses within a farm.

Section 3.4 (Cross-Farm Campylobacter Types):

  • At the beginning of the subsection: “Here, we investigate cgMLST types that appeared in more than one farm, suggesting cross-farm contamination or a shared external source.”
  • At the end of the subsection: “The broad distribution and recurrence of certain cross-farm types make them epidemiologically important, and their overlap with human cases is further discussed in Section 3.7.

Section 3.5 (Boot Socks vs. Cloacal Swabs):

  • At the beginning of the subsection: “This section compares cgMLST types detected in environmental samples (boot socks) 7-10 days prior to slaughter with those found in broiler samples (cloacal swabs), to evaluate the representativeness and added value of environmental sampling.”

  • At the end of the subsection: “These observations show that boot sock sampling of the broiler houses 7-10 days before slaughter can capture environmental genotypes that may not be detected in broilers at slaughter, highlighting its value for early detection and surveillance.”

Section 3.6 (Environmental Detection and Additional Sampling):

  • At the beginning of the subsection: “Here, we report findings from samples collected outside broiler houses, including manure, flies, and other environmental sources, to assess whether external reservoirs contribute to Campylobacter presence on farms.”

  • At the end of the subsection: “"These findings complete the overview of environmental Campylobacter detections outside broiler houses, highlighting that most cgMLST types identified in the external environment were not detected in animals or housing samples.

Section 3.7 (Overlap with Human Infections):

  • At the beginning of the subsection: In the final section, we examine the overlap between cgMLST types detected in broilers and those observed in the concurrent Danish human surveillance data. This comparison allows assessment of potential public health relevance.”

  • At the end of the subsection: “The high proportion of overlapping cgMLST types—particularly among cross-farm and persistent types—highlights their zoonotic potential and underscores the importance of continued genomic surveillance for public health.”

We believe the revised Results section now offers a clearer, more intuitive narrative, guiding the reader step-by-step through the study findings.

Comments 5: Discussion section: The discussion is fragmented into several subsections, each addressing limited aspects with insufficient depth. I recommend reorganizing this section into a single, cohesive narrative, allowing for more comprehensive exploration of the findings and better integration with the literature.

Response 5: We thank the reviewer for this valuable and constructive comment. We have substantially revised the entire discussion section to improve narrative flow, reduce fragmentation, and enable deeper synthesis of the study’s key findings. Rather than addressing isolated aspects across multiple shorter subsections, we now present the discussion in four integrated thematic sections, each of which links closely with the study objectives, public health context, and relevant literature:

  • 4.1 Strain Diversity and Persistence
  • Zoonotic Overlap and Transmission Pathways
  • Environmental Monitoring and Early Detection
  • Implications, Limitations, and Future Directions

We hope the revised structure better supports the interpretive depth and integration the reviewer requested.

Comments 6: Recommendations for control: The authors should expand the discussion to include practical implications for Campylobacter control. Based on the findings, what strategies can be proposed to improve prevention and control measures on broiler farms? How can this data inform surveillance or biosecurity practices?

Response 6:  We thank the reviewer for highlighting the need to elaborate practical implications for Campylobacter control. We have now expanded the discussion to explicitly address how our findings can inform prevention, control, and surveillance strategies in broiler production. Specifically:

In Section 4.2 (Zoonotic Overlap and Transmission Pathways), we emphasize the need to:

Prioritize farms with persistent and cross-farm genotypes for intensified monitoring

Enhance biosecurity in conventional systems, where high-risk strains were more prevalent

Investigate upstream transmission sources such as hatcheries and parent flocks

In Section 4.3 (Environmental Monitoring and Early Detection), we demonstrate the value of:

Boot sock sampling and environmental testing as early warning tools

Inclusion of non-poultry reservoirs (e.g. neighboring livestock) in surveillance efforts

Integration of environmental sampling in routine monitoring protocols to identify risk before colonization

In Section 4.4 (Implications, Limitations and Future Directions), we outline broader recommendations, including:

Shifting toward integrated, longitudinal, and genomics-informed surveillance strategies

Incorporating culture-independent methods

Prioritizing high-risk strains and production sites for targeted intervention

We believe these additions clearly outline the real-world applications of our findings and directly address the reviewer’s request for actionable strategies to enhance control and surveillance.

Comments 7: Ethical approval: The manuscript does not include the name of the ethics committee nor the corresponding approval number. This information is essential for studies involving animal sampling and must be clearly stated in accordance with the journal’s requirements and international ethical standards. The authors should include these details in a dedicated section at the end of the Materials and Methods, as specified in the journal’s guidelines.

Response 7: Thank you for your comment. We understand the importance of clearly stating ethical considerations related to animal sampling. We would like to clarify that this study did not require approval from an ethics committee, as all sampling was conducted on broiler carcasses post-mortem at slaughterhouses and did not involve any procedures on live animals. According to Danish legislation and ethical standards, such sampling does not fall within the scope of animal experimentation requiring ethical approval. To address your concern and align with the journal’s guidelines, we have moved the ethical information to the Institutional Review Board Statement section, which now reads: Sampling of broiler carcasses at slaughterhouses was conducted in collaboration with industry partners and complied with national legislation. As the study did not involve live animals or animal experimentation, ethical approval was not required under Danish law.
We hope this clarification satisfies the journal’s requirements regarding ethical transparency.

Minor Remarks

  • Comments 8: Line 21: Remove italics from the term campylobacteriosis.
    Response 8: This has been corrected.
  • Comments 9: Keywords: Eliminate any terms already present in the title and ensure the number of keywords adheres to journal guidelines.
    Response 9: Thank you for pointing this out. We have adapted the keywords as suggested, now including these: Whole-genome sequencing, Genomic surveillance, Longitudinal strain Persistence, Inter-farm transmission, Zoonosis, One Health, Foodborne pathogens

  • Comments 10: Lines 53–55: This paragraph is too short and should be merged with adjacent text.
    Response 10: This has been corrected.
  • Comments 11: Lines 74–84: These lines should be consolidated into a single paragraph for better flow.
    Response 11: This is now corrected.
  • Comments 12: Introduction: It is important to clarify two key points: (i) the main Campylobacterspecies involved in foodborne illnesses, and (ii) that poultry commonly serve as asymptomatic carriers of this pathogen.

    Response 12: We appreciate the reviewer’s comment and have clarified both points in the revised introduction. The following sentence has been added: Of the many species within the Campylobacter genus, Campylobacter jejuni and Campylobacter coli are the most commonly associated with human disease. C. jejuni ac-counts for the vast majority of human gastroenteritis cases globally and in Denmark, while C. coli is less prevalent but still clinically relevant [1]. Both species are found in poultry, but C. jejuni is the dominant species in broiler chickens [4]. In animals, these species are typically asymptomatic colonizers, yet their zoonotic potential poses a significant food safety concern [1].

This addition outlines the primary Campylobacter species implicated in foodborne illness and emphasizes the role of poultry as asymptomatic reservoirs, thereby addressing both aspects raised by the reviewer.

  • Comments 13: Lines 91–93: The relevance of this information is unclear; please clarify or remove.
    Response 13: This has been removed

  • Comments 14: Line 87: Verify numerical consistency—26 farms are mentioned in the abstract, but this is not clearly reflected in the main text. Although this number appears in Figure 1, the sampling design across farms and slaughterhouses must be explained more clearly in the Methods section.

Response 14: Thank you for pointing this out. To ensure consistency with the Abstract and Figure 1, we have explicitly stated the total number of farms (26) and their distribution by production type and year in the first paragraph of Section 2.1. The revised paragraph now reads:
This study included a total of 26 unique broiler farms, comprising 17 conventional farms and 9 free-range farms, sampled during 2020 and 2021. Of these, 15 conventional and 2 free-range farms were included in 2020. In 2021, seven new free-range farms (F4 to F10) and two new conventional farms (C15 and C17) were added, and seven farms were sampled in both years (C2, C4, C7, C8, C10, C12, and C14).
We believe these clarifications improve the transparency and coherence of the study design description in the Methods section.

  • Comments 15: Section 2.2: Rename this section to “Microbiological Analyses.” The isolation method used should be properly cited.
    Response 15: Thank you for your suggestion. We have renamed Section 2.2 to Microbiological Analyses as recommended. In addition, we have added that the culturing process was based on a modified version of ISO 10272-1:2017.

  • Comments 16: Line 134: Provide details on the presumptive differences used for species characterization.
    Response 16: We have added that it was by MALDI-TOF to the sentence: For broiler flocks that tested positive for Campylobacter, three representative isolates of jejuni (if available) and one isolate of C. coli (if detected), as determined by MALDI-TOF, underwent whole-genome sequencing (WGS).

  • Comments 17: Section 2.4: This section should be relocated to the end of the manuscript as per the journal’s submission guidelines. Include the name of the ethics committee and approval number.

Response 17: Thank you for your comment. As noted in our response to your earlier remark regarding ethical approval, this study did not require approval from an ethics committee, as all sampling was conducted post-mortem on broiler carcasses and did not involve any procedures on live animals. Under Danish legislation, such sampling is not classified as animal experimentation and does not require ethics committee review or an approval number. In line with the journal’s submission guidelines, we have now relocated the ethical statement to the Institutional Review Board Statement section at the end of the manuscript.

We hope this resolves the issue and meets the journal’s requirements for ethical reporting.

Comments 18: Throughout the manuscript and figures, replace generic references to “type” with the correct notation for sequence types (ST).
Response 18: Thank you for your comment. We agree that precise terminology is essential when describing sequence classifications. In this study, we used core genome multilocus sequence typing (cgMLST), which groups isolates based on allelic differences across 1,343 loci. Unlike traditional 7-locus MLST, cgMLST-based clusters do not have standardized sequence type (ST) numbers, and therefore cannot be referred to by universal ST designations. For clarity, we have defined the term “cgMLST type” at its first mention in the manuscript and ensured consistent use of this term throughout the text and figure legends (using simply “type” only where cgMLST is clearly defined in context). This approach accurately reflects the methodology applied and avoids confusion with classical MLST-based ST nomenclature.

  • Comments 19: Line 402: Reconsider the use of the term “vectors”, which typically refers to invertebrate organisms; use more appropriate terminology if referring to vehicles of transmission.

Response 19: Thank you for pointing this out. We agree and no longer use this term.

  • Comments 20: Line 414: Provide references for the outbreaks mentioned.
    Response 20: We have now clarified the source of the outbreak information in the manuscript and added references to the Danish Annual Reports on Zoonoses for the relevant years (Refs. 23 and 24). The text now specifies that the outbreaks were identified through national surveillance data and outbreak investigations conducted by Statens Serum Institut, with some of these also partially reported in the cited Annual Reports.

Comments 21: Section 4.5: Include proper references to support the content presented.
 Response 21: Thank you for pointing this out. We have added the following five references to the section. Note that due to your previous comment on rearrangement of the discussion, the section is no longer 4.5 and has now been combined with other parts. But highlighted here are the references we inserted to support your comment:

4.3 Environmental Monitoring and Early Detection

Environmental sampling played a key role in detecting Campylobacter cgMLST types that were not identified in broiler samples.

Although boot sock sampling did not capture the full diversity of cgMLST types found in cloacal swabs, it consistently detected the dominant genotype in most flocks that were positive in both methods. Differences between the two sampling approaches may reflect both variation in sampling timing and variation in the number of isolates sequenced from each sample type. In some cases, cgMLST types were detected in boot socks but not in cloacal swabs, indicating that environmental contamination may precede or persist beyond detectable colonization in broilers. For example, the cgMLST type ST7355-1 was identified in boot socks without being present in cloacal swabs from the same flock, pointing to environmental reservoirs as potential sources of delayed or intermittent recontamination of flocks. Together, these findings highlight that environmental detection can serve as an early indicator of risk and support the utility of boot socks as a practical early-warning tool for detecting Campylobacter in the broiler house environment [26].

Sampling outside broiler houses resulted in a low detection rate, but this likely reflects limitations in sensitivity or sample coverage rather than a true absence of Campylobacter [27]. Notably, in our study Campylobacter was identified in cattle and pig manure from nearby farms, indicating potential interspecies transmission or cross-farm contamination. This observation aligns with recent Danish WGS-based studies, where cattle were highlighted as a plausible reservoir for Campylobacter in poultry production, while pigs appeared to play a more limited role [15]. Although the exact role of non-poultry reservoirs remains unclear, their presence supports the need for broader environmental surveillance [4].

Collectively, these findings indicate that incorporating environmental sampling—both within and outside broiler houses—can enhance surveillance sensitivity and provide earlier signals of contamination. This is particularly relevant on farms where persistent or high-risk cgMLST types have been detected, as environmental reservoirs may facilitate continued transmission. Integrating such strategies into monitoring programs could help prevent flock colonization and reduce the likelihood of broader spread [28].

  1. (BIOHAZ), E.P. on B.H.; Koutsoumanis, K.; Allende, A.; Alvarez-Ordóñez, A.; Bolton, D.; Bover-Cid, S.; Davies, R.; De Cesare, A.; Herman, L.; Hilbert, F.; et al. Update and Review of Control Options for Campylobacter in Broilers at Primary Production. EFSA J. 2020, 18, e06090, doi:https://doi.org/10.2903/j.efsa.2020.6090.
  2. Lassen, B.; Takeuchi-Storm, N.; Henri, C.; Hald, T.; Sandberg, M.; Ellis-Iversen, J. Analysis of Reservoir Sources of Campylobacter Isolates to Free-Range Broilers in Denmark. Poult. Sci. 2023, 102, 103025, doi:10.1016/j.psj.2023.103025.
  3. Matt, M.; Nordentoft, S.; Kopacka, I.; Pölzler, T.; Lassnig, H.; Jelovcan, S.; Stüger, H.P. Estimating Sensitivity and Specificity of a PCR for Boot Socks to Detect Campylobacter in Broiler Primary Production Using Bayesian Latent Class Analysis. Prev. Vet. Med. 2016, 128, 51–57, doi:https://doi.org/10.1016/j.prevetmed.2016.03.015.
  4. Adkin, A.; Hartnett, E.; Jordan, L.; Newell, D.; Davison, H. Use of a Systematic Review to Assist the Development of Campylobacter Control Strategies in Broilers. J. Appl. Microbiol. 2006, 100, 306–315, doi:10.1111/j.1365-2672.2005.02781.x.
  5. Llarena, A.-K.; Skjerve, E.; Bjørkøy, S.; Forseth, M.; Winge, J.; Hauge, S.J.; Johannessen, G.S.; Spilsberg, B.; Nagel-Alne, G.E. Rapid Detection of Campylobacter Spp. in Chickens before Slaughter. Food Microbiol. 2022, 103, 103949, doi:10.1016/j.fm.2021.103949.

Comments 22: Improve all table titles to ensure they are self-explanatory and understandable without reference to the main text.

Response 22: We thank the reviewer for this helpful suggestion. We have revised the titles and legends of all tables and figures to ensure that they are fully self-explanatory and can be understood without referring to the main text. This includes expanding the titles to clearly describe the content and scope of each table/figure, defining all abbreviations, and adding clarifying notes where necessary (e.g., category definitions, inclusion criteria, and data sources). These changes improve the readability and stand-alone interpretability of the tables and figures in the manuscript.

Reviewer 3 Report

Comments and Suggestions for Authors

The manuscript “Clonal Spread and Persistence of Campylobacter in Danish Broiler Farms and Association with Human Infections” by Katrine Grimstrup Joensen and coauthors conducted a WGS study on farm samples to identify Campylobacter diversity based on STs and compared the spatial and temporal flow between flocks. The study shows a considerable sampling and offers important information that could contribute to understanding the Campylobacter dynamics. However, there are some observations that could improve the manuscript.
•    Major observation: It might have been more accurate to focus solely on C. jejuni. In any case, it would be better if the authors presented the results (and discussions) separately by species.
•    Introduction: I suggest including a brief description and differentiation of the main species of the Campylobacter genus, highlighting their clinical (animal and human) and epidemiological significances.
•    Lines 78-83: These lines are discussions. Move and/or merge them into their respective sections.
•    Section 2.1: I suggest starting the description by mentioning the total number of farms used (17 conventional, 10 free-range). After this, continue with the periods and details as previously described.
•    Lines 146-147: Detail the tool and version used to determine the MLST profile.
•    Lines 162-163: Describe the strategy or method used for species differentiation. Also, provide more detailed details on the quantity and percentage of each species. Preferably, also indicate the change in these percentages by year.
•    Lines 172-176: Authors should include the definition of a Campylobacter-positive flock. Is it based on MLSTs or another genomic marker or tool? Furthermore, does it imply the presence of any of the three Campylobacter species?
•    Line 237: Did you mean Table S1?
•    Line 246: Appendix 2 was not available.
•    Section 3.5: The authors mention that 12 of the 18 species detected in the boot sock samples were also detected in swabs. Please also indicate how many were detected in the swabs and how many were not detected in the boot sock samples. If the diversity detected in sewage swabs is much greater than that in the boot sock samples, the statement in lines 264-265 seems overstated. In that case, I suggest changing to "partially mirrored."
•    Discussion: I suggest adding a minimum spanning tree or similar analysis to analyze the proportion and distribution of clonal complexes. Discuss possible clusters and related CCs.
•    Lines 425-427: This statement seems overstated. Defining it as "closely" would imply a bidirectional proximity (in both sample types) of all detected types. However, the authors did not categorically show this unique and shared diversity in terms of percentages.
•    Lines 434-437: Compare and discuss these findings with references from other studies.

Author Response

Comments 1: The manuscript “Clonal Spread and Persistence of Campylobacter in Danish Broiler Farms and Association with Human Infections” by Katrine Grimstrup Joensen and coauthors conducted a WGS study on farm samples to identify Campylobacter diversity based on STs and compared the spatial and temporal flow between flocks. The study shows a considerable sampling and offers important information that could contribute to understanding the Campylobacter dynamics. However, there are some observations that could improve the manuscript.

Response 1: We greatly appreciate the reviewer’s positive overall assessment of our study’s scope, sampling effort, and contribution to understanding Campylobacter dynamics. We also value the constructive suggestions for clarifying species presentation, refining methodological descriptions, and improving the integration of results with the epidemiological context. In the revisions, we have addressed the reviewer’s main concerns, including clarifying species distribution, refining definitions and methodological details, and adjusting interpretations to better reflect the data. Where we chose an alternative approach, we have explained our rationale in the detailed responses below. We believe these revisions substantially strengthen the clarity, focus, and scientific impact of the manuscript.

Comments 2: Major observation: It might have been more accurate to focus solely on C. jejuni. In any case, it would be better if the authors presented the results (and discussions) separately by species.

Response 2: We acknowledge the reviewer’s suggestion to present results separately by species. However, we chose to analyze C. jejuni and C. coli together for several reasons. First, although C. jejuni is the dominant species in Danish broiler production, C. coli—while less prevalent—remains clinically relevant and was detected in our dataset. Both species share similar transmission routes within poultry production systems, and both have zoonotic potential, with isolates from each species matching human cases in our surveillance data. Presenting them jointly reflects the integrated nature of transmission risk in a One Health context and allows us to identify contamination patterns, persistence, and cross-farm spread that apply to the whole Campylobacter population. Second, separating the results would have substantially increased the complexity and length of the manuscript without meaningfully changing the main epidemiological conclusions, since our findings and recommendations (e.g., regarding persistent and cross-farm types, environmental reservoirs, and surveillance priorities) apply to both species. Finally, combining the species in the main analyses is consistent with our aim to provide an integrated, One Health–oriented view of Campylobacter epidemiology in Danish broiler production.

Comments 3: Introduction: I suggest including a brief description and differentiation of the main species of the Campylobacter genus, highlighting their clinical (animal and human) and epidemiological significances.

Response 3: Thank you for the suggestion. To improve clarity and context, we have added a short paragraph early in the Introduction that describes the most clinically relevant Campylobacter species (C. jejuni and C. coli), including their relevance for human disease and their role as asymptomatic colonizers in poultry. The following paragraph has been inserted: Of the many species within the Campylobacter genus, Campylobacter jejuni and Campylobacter coli are the most commonly associated with human disease. C. jejuni accounts for the vast majority of human gastroenteritis cases globally and in Denmark, while C. coli is less prevalent but still clinically relevant. Both species are found in poultry, but C. jejuni is the dominant species in broiler chickens. In animals, these species are typically asymptomatic colonizers, yet their zoonotic potential poses a significant food safety concern.
We believe this addition strengthens the epidemiological framing of the study.

Comments 4: Lines 78-83: These lines are discussions. Move and/or merge them into their respective sections.
Response 4: Thank you for pointing this out. We agree that the original phrasing of the final lines in the Introduction had a forward-looking tone more typical of a discussion or conclusion. To address this, we have rephrased the sentences to better reflect the study’s objectives and rationale, ensuring they align with the informative and contextual tone of the Introduction. The revised paragraph now reads: This study aims to generate high-resolution genomic data on Campylobacter populations circulating in Danish broiler production. Such knowledge is essential to inform targeted control strategies and ultimately contribute to reducing the risk of Campylobacter-related human infections through improved surveillance and intervention planning.

Comments 5: Section 2.1: I suggest starting the description by mentioning the total number of farms used (17 conventional, 10 free-range). After this, continue with the periods and details as previously described.

Response 4: Thank you for this helpful suggestion. We have revised Section 2.1 to provide a clearer overview of the study design. The section now opens by stating the total number and type of broiler farms included (17 conventional and 9 free-range) and their distribution across the two study years. This restructuring improves clarity for the reader and aligns with your recommendation. Specifically, the first paragraph of Section 2.1 now reads: This study included a total of 26 unique broiler farms, comprising 17 conventional farms and 9 free-range farms, sampled during 2020 and 2021. Of these, 15 conventional and 2 free-range farms were included in 2020. In 2021, seven new free-range farms (F4 to F10) and two new conventional farms (C15 and C17) were added, and seven farms were sampled in both years (C2, C4, C7, C8, C10, C12, and C14).
We hope this restructuring addresses your comment and improves the readability of the study design.

Comments 6: Lines 146-147: Detail the tool and version used to determine the MLST profile.
Response 6: Thank you for the comment. We have clarified in the Methods section that MLST profiles were primarily determined using the MLST module in the Bifrost pipeline, as previously described. In cases where Bifrost did not assign a sequence type, MLST analysis was instead performed using BioNumerics version 8.1 (BioMérieux). We have now included this additional detail in the relevant sentence in the Methods section to improve transparency and reproducibility.

Comments 7: Lines 162-163: Describe the strategy or method used for species differentiation. Also, provide more detailed details on the quantity and percentage of each species. Preferably, also indicate the change in these percentages by year.
Response 7: We thank the reviewer for this comment. As requested, we have now clarified the method for species identification by explicitly stating that all positive samples were characterized using MALDI-TOF, as detailed in Section 2.2. Additionally, we have included specific numbers and proportions of each Campylobacter species detected (C. jejuni, C. coli, and C. lari) for both study years. This information is now presented in Section 3.1, and the species distribution is broken down by year to show temporal variation.

Comments 8: Lines 172-176: Authors should include the definition of a Campylobacter-positive flock. Is it based on MLSTs or another genomic marker or tool? Furthermore, does it imply the presence of any of the three Campylobacter species?
Response 8: We thank the reviewer for this relevant comment. As suggested, we have clarified the definition of a Campylobacter-positive flock in Section 2.2 of the Methods: “A broiler flock was considered positive for Campylobacter if one or more of the ten samples were culture-positive. Results from cloacal swabs and boot socks were considered jointly when determining flock status, as both sample types represent fecal material from the same birds. The presence of any of the three targeted species (C. jejuni, C. coli, or C. lari) qualified the flock as positive.”

This classification was based on conventional culturing followed by species determination using MALDI-TOF, and not on MLST or other genomic markers. In the Results section, we have also retained a brief statement of this definition for clarity: “As defined in the Methods, a flock was considered Campylobacter-positive if at least one out of ten collected cloacal swab samples or one boot sock sample from inside the house was culture-positive.”
Comments 9: Line 237: Did you mean Table S1?
Response 9: Thank you for this observation. This has now been corrected.
Comments 10: Line 246: Appendix 2 was not available.
Response 10: You are right. We decided not to include this Appendix 2 in the final manuscript as it was basically a condensed version of the information in Table S1. Thus, we have now deleted the sentence “Details of these types are provided in Appendix 2”.

Comments 11: Section 3.5: The authors mention that 12 of the 18 species detected in the boot sock samples were also detected in swabs. Please also indicate how many were detected in the swabs and how many were not detected in the boot sock samples. If the diversity detected in sewage swabs is much greater than that in the boot sock samples, the statement in lines 264-265 seems overstated. In that case, I suggest changing to "partially mirrored."

Response 11: We thank the reviewer for this helpful comment. In response, we have revised Section 3.5 to clarify the overall diversity observed in each sample type and to more accurately describe the degree of overlap between boot sock and cloacal swab results. Specifically, we now state that a total of 18 different cgMLST types were detected across all boot sock samples collected in the study. In the subset of 19 flocks that were positive in both sample types and could be directly compared, a higher number of cgMLST types was observed in cloacal swabs than in boot socks, and several types were detected only in swabs. Accordingly, we have moderated our interpretation of the overlap between methods. The original statement referring to a “close correspondence” was replaced with: This indicates a partial correspondence, where boot sock sampling mirrored some -but not all - types found in cloacal swabs from the same flocks.

This revised wording better reflects the observed data and aligns with the reviewer’s suggestion.

Comments 12: Discussion: I suggest adding a minimum spanning tree or similar analysis to analyze the proportion and distribution of clonal complexes. Discuss possible clusters and related CCs.
Response 12: We thank the reviewer for this suggestion. However, our study is based on high-resolution core genome MLST (cgMLST) analysis, which provides a much finer-scale view of genetic relationships than traditional MLST-based clonal complex (CC) groupings. In this context, clonal complexes were not central to our analytical framework, and we did not consider them essential for interpreting our results. Therefore, we have chosen not to include a minimum spanning tree or specific analyses of CC proportion and distribution. Instead, we focused on cgMLST allelic differences and genome-wide clustering, which offer a more detailed understanding of isolate relatedness in line with the study’s objectives.

Comments 13: Lines 425-427: This statement seems overstated. Defining it as "closely" would imply a bidirectional proximity (in both sample types) of all detected types. However, the authors did not categorically show this unique and shared diversity in terms of percentages.

Response 13: We thank the reviewer for this helpful clarification. In response, we have revised both the wording and placement of the statement in the updated Discussion. The previous phrasing (“correlated closely”) has been replaced with a more precise description that reflects the observed patterns. The revised sentence now reads: Although boot sock sampling did not capture the full diversity of cgMLST types found in cloacal swabs, it consistently detected the dominant genotype in most flocks that were positive in both methods.
This revised wording avoids overstatement and more accurately reflects the partial, but epidemiologically relevant, overlap observed in our data. The sentence now appears in Section 4.3 Environmental Monitoring and Early Detection.

Comments 14: Lines 434-437: Compare and discuss these findings with references from other studies.
Response 14: Thank you for your comment. We have revised this section of the Discussion to include direct comparison with recent Danish WGS-based studies addressing non-poultry reservoirs, and have added supporting references. The updated text now reads: Sampling outside broiler houses resulted in a low detection rate, but this likely reflects limitations in sensitivity or sample coverage rather than a true absence of Campylobacter [27]. Notably, in our study Campylobacter was identified in cattle and pig manure from nearby farms, indicating potential interspecies transmission or cross-farm contamination. This observation aligns with recent Danish WGS-based studies, where cattle were highlighted as a plausible reservoir for Campylobacter in poultry production, while pigs appeared to play a more limited role [15]. Although the exact role of non-poultry reservoirs remains unclear, their presence supports the need for broader environmental surveillance [4].
This addition strengthens the contextualization of our findings and directly addresses the reviewer’s request for comparison with existing literature.

Round 2

Reviewer 2 Report

Comments and Suggestions for Authors

All my suggestions were adressed and the manuscript was improved.

Reviewer 3 Report

Comments and Suggestions for Authors

The authors have appropriately justified or responded to the observations made previously.